

# A novel automated Parkinson's disease identification approach using deep learning and EEG

Marwa Obayya[1], Muhammad Kashif Saeed[2], Mashael Maashi[3], Saud S. Alotaibi[4], Ahmed S. Salama[5] and Manar Ahmed Hamza[6]

[1] Department of Biomedical Engineering, College of Engineering, Princess Nourah bint Abdulrahman University, Riyadh, Saudi Arabia
[2] Department of Computer Science, King Khalid University, Abha, Saudi Arabia
[3] Department of Software Engineering, College of Computer and Information Sciences, King Saud University, Riyadh, Saudi Arabia
[4] Department of Information Systems, Umm Al-Qura University, Mecca, Saudi Arabia
[5] Department of Electrical Engineering, Future University in Egypt, New Cairo, New Cairo, Egypt
[6] Department of Computer and Self Development, Prince Sattam bin Abdulaziz University, AlKharj, Saudi Arabia

Corresponding author
Muhammad Kashif Saeed,
mksaeed@kku.edu.sa

## ABSTRACT

The neurological ailment known as Parkinson's disease (PD) affects people throughout the globe. The neurodegenerative PD-related disorder primarily affects people in middle to late life. Motor symptoms such as tremors, muscle rigidity, and sluggish, clumsy movement are common in patients with this disorder. Genetic and environmental variables play significant roles in the development of PD. Despite much investigation, the root cause of this neurodegenerative disease is still unidentified. Clinical diagnostics rely heavily on promptly detecting such irregularities to slow or stop the progression of illnesses successfully. Because of its direct correlation with brain activity, electroencephalography (EEG) is an essential PD diagnostic technique. Electroencephalography, or EEG, data are biomarkers of brain activity changes. However, these signals are non-linear, non-stationary, and complicated, making analysis difficult. One must often resort to a lengthy human labor process to accomplish results using traditional machine-learning approaches. The breakdown, feature extraction, and classification processes are typical examples of these stages. To overcome these obstacles, we present a novel deep-learning model for the automated identification of Parkinson's disease (PD). The Gabor transform, a standard method in EEG signal processing, was used to turn the raw data from the EEG recordings into spectrograms. In this research, we propose densely linked bidirectional long short-term memory (DLBLSTM), which first represents each layer as the sum of its hidden state plus the hidden states of all layers above it, then recursively transmits that representation to all layers below it. This study's suggested deep learning model was trained using these spectrograms as input data. Using a robust sixfold cross-validation method, the proposed model showed excellent accuracy with a classification accuracy of 99.6%. The results indicate that the suggested algorithm can automatically identify PD.

## INTRODUCTION

As individuals age, there is a notable decline in the quantity of connections between brain cells, accompanied by a reduction in the size of neurons. In contrast to muscle, skin, and bone cells, nerve cells exhibit limited regenerative capacity. According to previous research, it has been observed that as individuals progress in age, there is a notable occurrence of neuronal death or impairment (*Department Human Services & National Health, 2001*). PD is a widely recognized neurodegenerative disorder defined by the gradual deterioration of neurons in the substantia nigra region of the brain. This condition leads to the impairment of motor functions and the progress of various symptoms related to movement difficulties. The underlying cause of PD is the damage and loss of these specific neurons, which produce dopamine, a neurotransmitter crucial for regulating movement and coordination. The disruption of typical basal ganglia operating caused by the degeneration of these neurons is responsible for the motor indications often seen in individuals with PD. The neurons in question are accountable for synthesizing and releasing a neurotransmitter called dopamine. Dopamine, a neurotransmitter, is a crucial chemical mediator in facilitating communication between neurons within the brain. The brain plays a vital role in facilitating communication between different body regions, ensuring their proper functioning. This is particularly evident in the coordination of body movements and speech delivery. PD symptoms manifest when there is a significant loss of dopaminergic neurons or an abnormal level of dopamine in the brain (*Poewe et al., 2017*).

Electroencephalography (EEG) is a standard neuroimaging procedure that involves attaching electrodes to a subject's scalp to record and analyze the brain's electrical activity. These electrodes record voltages generated by local neuronal activity in the brain. Researchers can learn about several facets of brain activity and function by analyzing these voltage patterns. The EEG has been an invaluable resource for researchers in cognitive neuroscience, sleep science, and neurological disease. Epilepsy, sleep problems, and sensory transmission are only a few examples of the many ailments for which non-invasive electronic devices have been extensively employed in medical evaluation (*Michel & Brunet, 2019*). Skilled neurophysiologists can subjectively and relatively assess and understand the underlying medical states thanks to observable and substantial changes in EEG patterns across various disorders. Some neurological disorders have been noted to have subtler shifts. However, due to the complexity of EEG signals, it can be difficult for human observers and existing evaluation methods to determine the significance of these shifts reliably. Diagnosing Alzheimer's, brain tumors, and PD with EEG has shown much promise, and there has been an increased interest in research into the efficacy of deep learning (DL) and neural networks in this regard (*Cecere, Corrado & Polikar, 2014*; *Liu et al., 2020*).

In the non-automated diagnosis of PD using EEG signals, a standard procedure typically involves several key steps. Firstly, electrodes are placed on the patient's scalp to record EEG data, capturing the brain's electrical activity. The recorded EEG signals are then pre-processed, which includes tasks such as filtering to remove noise and artifacts, segmentation to isolate relevant portions of the data, and feature extraction to derive meaningful information from the EEG signals, such as spectral power or coherence measures.

These features are subsequently analyzed by medical professionals, often neurologists or neurophysiologists, who look for distinctive patterns or abnormalities indicative of PD. The diagnosis is then made based on their clinical expertise and observations. Challenges in this non-automated approach include subjectivity in interpretation, as different experts may reach different conclusions from the same EEG data. It also relies heavily on the experience and training of the clinician, which can introduce variability in diagnostic accuracy.

Moreover, manual diagnosis can be time-consuming and may not leverage the full potential of advanced signal processing and machine learning techniques that could provide more objective and automated diagnostic aids. However, the advantage of this approach is that it can benefit from the expertise of skilled clinicians who can integrate EEG findings with other clinical assessments to make a comprehensive diagnosis. Nevertheless, the field is increasingly exploring incorporating automated methods and machine learning algorithms to augment the diagnostic process, potentially improving accuracy and efficiency.

The use of machine and deep learning methods for the automated diagnosis and categorization of PD has gained popularity in the past few years. These approaches utilize different data sources, such as EEG, Magnetic Resonance Imaging (MRI), patterns of speech, written exams, interactions, and sensory data. These diverse data sources allow for a comprehensive analysis of PD, enabling more accurate detection and categorization of the disease. Recent studies have explored various modalities to implement deep learning techniques in detecting Parkinson's disease. However, the current clinical diagnosis still heavily relies on the anomalies in the motor system observed, which is subjective and susceptible to human error. Moreover, it is worth noting that there is currently a lack of distinctive or well-established clinical biomarkers associated with the disease or its related issues.

In machine learning, conventional methodologies have traditionally depended on labor-intensive procedures. These procedures encompass various stages, such as signal decomposition, feature extraction, and classification, all necessary to address the intricate nature of EEG signals. To address the limitations mentioned earlier, a novel and pioneering deep-learning model has been developed to facilitate the automated detection and identification of Parkinson's disease. This study utilized the Gabor transform, a widely used technique in EEG signal processing, to transform the raw EEG data into spectrograms. The DLBLSTM model is a proposed manner that exhibits a distinctive structure. In this model, each layer is characterized by aggregating its hidden state and the hidden states of all the layers positioned above it. This representation is then recursively transmitted to all the layers positioned below it.

The remainder of this paper is organized in the following manner: The second section thoroughly analyzes the relevant literature. 'Methods and Materials' details the dataset utilized in this study, including the preprocessing and preparation techniques employed to enhance the quality and representation of the EEG signals. 'Result and Discussion' of the research paper presents the results obtained from the suggested approach. 'Conclusion and Future Work' of this research paper offers the conclusion, highlighting the various contributions made by this study and discussing its possible uses in real-world scenarios.

# RELATED WORKS

*Zhan et al. (2018)* employed machine-learning techniques and smartphone-based tasks to evaluate the level of daily symptoms associated with PD. This approach led to the development of the mobile PD score. This study conducted a comprehensive evaluation on a sample size of 129 individuals. The researchers found that gait, among other factors, played a significant role in determining the overall score. Specifically, gait was found to have the highest contribution. The results demonstrated a strong correlation between the newly developed and existing scales commonly used for in-person evaluations, including the PD Scale of the Society of Total Disorders, the Timed Up and test, and the Hoehn and Yahr stage assessment. According to *Okuma et al. (2018)*, falls are considered to be the loss of independence, one of the worst outcomes of PD.

In a study by *Gao et al. (2018)*, an investigation was carried out to evaluate the danger of falls in patients with PD using clinical, demographic, and neuroimaging data. In their study, the researchers employed various features to assess the participants. Features including gait speed, unstable posture, and measurements linked to gait difficulties were considered. In this study, the researchers examined the classification of two distinct classes: fall and no fall. To achieve this, they employed support vector machines (SVM) as their chosen classification algorithm. The results obtained from their experiments demonstrated accuracies reaching up to 83%. Speech disturbance is a notable characteristic observed in individuals diagnosed with PD. The database of voice records from people with PD and people in good health was used in a study by *Mostafa et al. (2019)*.

In their study, *Zhang (2017)* proposed the utilization of stack autoencoders (SAE) as a method for diagnosing PD remotely by complete telephone-based assessments. The researchers collected participants' personal information and vocal data, which were then inputted into a machine learning algorithm to analyze the speech records (*Zhang, 2017*). In a study by *Wagh & Varatharajah (2020)*, a novel 8-layer graph-CNN architecture was introduced to classify neurological diseases. The proposed model attained an impressive accuracy rate of 85% in accurately identifying and classifying these diseases. Researchers (*Koch et al., 2019*) explored using a Random Forest Classifier to identify PD. The classifier was intended to use EEG information manually and automatically retrieved by clinicians. The area under the receiver operating characteristic curve (AUC) analysis revealed that the suggested classifier achieved a 91% accuracy (*Koch et al., 2019*). Two hybrid models, the convolutional neural network (CNN) and a recurrent neural network (RNN), were presented by *Shi et al. (2019)*, which reported that the former model demonstrated a detection accuracy of 82.89% in identifying PD.

In a previous study by *Lee, Hussein & McKeown (2019)*, a hybrid model was introduced that effectively combined CNN and LSTM to leverage EEG data's spatial and temporal features. The projected model attained an impressive accuracy of 96.9% in accurately distinguishing between individuals with PD and healthy controls (HC). This study demonstrates modern machine learning methods' promise to enhance PD diagnostic precision and throughput. The model's learning process involves acquiring representations that strongly correlate with clinical features, specifically disease severity, and levels of

dopaminergic activity. Previous studies used a framework based on artificial neural networks (ANN) to analyze EEG data. This framework has been employed to differentiate individuals with PD from control subjects. The results of this approach have established a high level of accuracy, with a classification accuracy of 98%. Additionally, the sensitivity of the framework, which refers to its ability to identify individuals with PD correctly, was found to be 97%. Moreover, the specificity of the framework, which indicates its capacity to identify control subjects accurately, was determined to be 100% (*Shaban, 2021*).

To diagnose PD, the research team led by Shail Raval considers all relevant factors. The symptoms were the primary focus of this research. These included things like stiffness, resting tremors, changes in voice, etc. For secure data transport, techniques like checking for duplicates and identifying faulty nodes are recommended. The proposed method successfully covers long transmission distances. The concept of retransmission is also supported (*Raval, Balar & Patel, 2020*). This research examines acoustic equipment voice input for PD prediction. Predicting PD in patients involves analyzing the speech patterns of various individuals. Multilayer perceptron and logistic regression (LR) frameworks were used to detect PD in a speech dataset (*Mei, Desrosiers & Frasnelli, 2021*).

The suggested frameworks by *Parajuli, Amara & Shaban (2023)* begin by categorizing the sleep EEG time series into three phases of sleep, further converting the segmented communication in the domain of time-frequency using the constant wavelet transform and the variational mode disintegration, and then implementing novel convolutional neural networks on the time-frequency illustrations. Additionally, the suggested deep-learning algorithms were utilized to display the traits that allowed for an accurate prediction of moderate cognitive decline in Parkinson's disease. In this article (*Shaban & Amara, 2022*), we provide a deep-learning method based on a just-proposed 20-layer convolutional neural network (CNN) used on the visual representation of the Wavelet domain of a resting-state EEG. The suggested way successfully identified PD and distinguished between participants with PD who were taking medication and subjects who were not. A deep learning-based model for the diagnosis of Parkinson's disease (PD) utilizing resting state electroencephalogram (EEG) signals is presented in this article (*Delfan et al., 2023*). The study aims to create an automated model to extract intricate hidden nonlinear characteristics from EEG and show how it may be applied to unobserved data.

A workable medical decision-making strategy that aids doctors in identifying patients with PD was recently developed by *Kaur, Aggarwal & Rani (2020)*. Therefore, several hyperparameters need to be tweaked and established to evaluate DL algorithms; this work presents a specific method for optimizing grid searches in design for constructing an improved DL algorithm to anticipate the early diagnosis of PD. Hyperparameters, efficiency, and optimization of the DL technique are all part of the grid-searching optimization approach. Classifying MR images of healthy controls and PD patients using the DL-NN model is the focus of the research of *Sivaranjini & Sujatha (2020)*. AlexNet, a CNN architecture, is used to improve PD detection. The MR image is evaluated to yield accuracy metrics, and a transfer-learned network is then trained on the data. *Quan, Ren & Luo (2021)* introduced a Bi-LSTM technique to diagnose PD using a speech signal's time-series dynamics. The amount of energy of the under-voiced to unvoiced segment

and onset to voiced segment transitions is used to assess the flexible speech characteristic. To improve the accuracy of FOG identification in a real-world home setting, *Sigcha et al. (2020)* suggested a unique technique using RNN. For their PD prediction work, *Leung et al. (2021)* concentrated on creating DL, an ensemble technique. The first part of the process involved feature extraction using DaTscan, while the second part involved feature extraction using medical assessments of motor symptoms. Initial baseline screening findings four years later were predicted using an ensemble of DNN models trained on subsets of the retrieved features.

Researchers are using a material known as a metal–organic framework (MOF) to capture hyperpolarized xenon (*Zeng et al., 2020*; *Zhang et al., 2023*) selectively. The research likely investigates how disruptions in calcium balance contribute to the development and progression of Parkinson's disease. It may also explore potential treatments or interventions to restore proper calcium regulation as a therapeutic approach (*Zhang et al., 2022a*; *Zhu et al., 2021*; *Shen et al., 2020*). This involves monitoring factors like the driver's eye movements, heart rate, steering behavior, or other data sources to identify fatigue-related patterns and features (*Wang et al., 2022a*; *Shen et al., 2023*). Neurogenesis generates new nerve cells, neurons, from neural stem cells or progenitor cells (*Zhang et al., 2022b*; *Sun et al., 2023*). In healthcare, this data might include patient vital signs, electrocardiogram (ECG) readings, or other types of time-series data (*Wang et al., 2022b*; *Shan et al., 2023*). It entails the application of a low electrical current in an alternating pattern to specific areas of the brain through electrodes placed on the scalp (*Huang et al., 2023*; *Xu et al., 2022*). This approach can be precious in IoT applications where real-time responsiveness and adaptability to changing conditions are critical, such as healthcare monitoring systems (*Cheng et al., 2016*). The method is designed to retrieve similar lung CT images from extensive databases efficiently (*Zhuang et al., 2022*; *Zhuang, Jiang & Xu, 2022*). It likely utilizes the ORB algorithm, which is employed to detect and describe critical features within the images (*Zhang et al., 2022c*). By iteratively improving the image reconstruction using differential sparse techniques, this approach aims to produce higher-quality CT images while minimizing radiation exposure (*Lu et al., 2023b*; *Lu et al., 2023a*). Such modeling can be indispensable for various applications in cardiology, including the study of cardiac function and the development of medical devices (*Liu et al., 2023*; *Hu et al., 2021*).

Existing approaches for diagnosing PD using EEG data have their own set of pros and cons. Traditional manual diagnosis by expert neurologists offers the advantage of clinical expertise and the ability to consider various clinical symptoms and medical history in conjunction with EEG data. However, it can be subjective, time-consuming, and prone to inter-rater variability. On the other hand, automated machine learning approaches provide the advantage of objectivity and the potential to process large datasets rapidly. They can extract intricate patterns from EEG signals that might be challenging for human observers to discern. However, they often require substantial amounts of labeled data for training, and the interpretability of their results can be limited. The dense connected Bi-LSTM architecture represents a promising approach for PD diagnosis using EEG data. It combines the strengths of bidirectional LSTM networks, capable of capturing temporal

dependencies in EEG signals in both forward and backward directions, with the benefits of dense connections that facilitate the flow of information across network layers. This architecture excels in learning complex patterns and temporal dependencies within EEG data, making it well-suited for PD diagnosis. By leveraging the recurrent nature of LSTM networks, it can model long-range dependencies and subtle changes in EEG signals over time, which are essential for detecting PD-related abnormalities.

# METHODS AND MATERIALS

## Problem formulation

Given a dataset of EEG recordings, the objective is to develop a novel automated identification approach for PD using DL techniques. The goal is to accurately classify individuals as PD-positive or PD-negative based on their EEG signals. Let $X$ be the dataset consisting of N EEG recordings, where each recording $x_i$ corresponds to the EEG signal of the ith individual. Each EEG recording is signified as a time series of $T$ data points, denoted as $x_i = (x_{i1}, x_{i2}, \ldots, x_{iT})$, where $x_{ij}$ represents the $j$th data point in the ith recording.

The objective is to train a deep-learning model to learn the mapping function $F$ that can automatically identify Parkinson's disease from the EEG recordings. The model takes the raw EEG data as input and outputs a classification label $y_i$, indicating whether the individual is PD-positive ($y_i = 1$) or PD-negative ($y_i = 0$). Formally, the problem can be represented as follows:

Given the dataset $X = \{(x_1, y_1), (x_2, y_2), \ldots, (x_N, y_N)\}$, where $x_i$ represents the EEG recording for the ith individual and $y_i$ represents the corresponding label indicating PD-positive or PD-negative, the objective is to find the optimum parameters $\theta$ of the deep-learning model F that minimize the classification error:

$$\theta^* = \frac{arg\ min}{\theta} \frac{1}{N} \sum_{i=1}^{N} L\big(F(x_i; \theta), y_i\big) \tag{1}$$

where $L(\cdot)$ is a suitable loss function, such as cross-entropy, and $F(x_i; \theta)$ represents the output of the deep-learning model with parameters $\theta$ for the input EEG recording $x_i$. The aim is to train the deep-learning model using the proposed densely linked bidirectional long short-term memory (DLBLSTM) architecture, which leverages the Gabor transform to convert the raw EEG data into spectrograms. The trained model should demonstrate excellent performance in terms of classification accuracy, as validated using a robust sixfold cross-validation method.

## Proposed methodology

The raw EEG data from the recordings is pre-processed to remove noise and artifacts. This step ensures the data is in a suitable format for further analysis. The Gabor transform, a widely used technique in EEG signal processing, is applied to the pre-processed EEG data. This transform converts the time-domain EEG signals into spectrograms, which capture the frequency content of the signals over time. From Fig. 1, the overall structure of the Parkinson's disease detection methodology is depicted. The figure visually represents the different components and their connections within the proposed approach. Figure 1

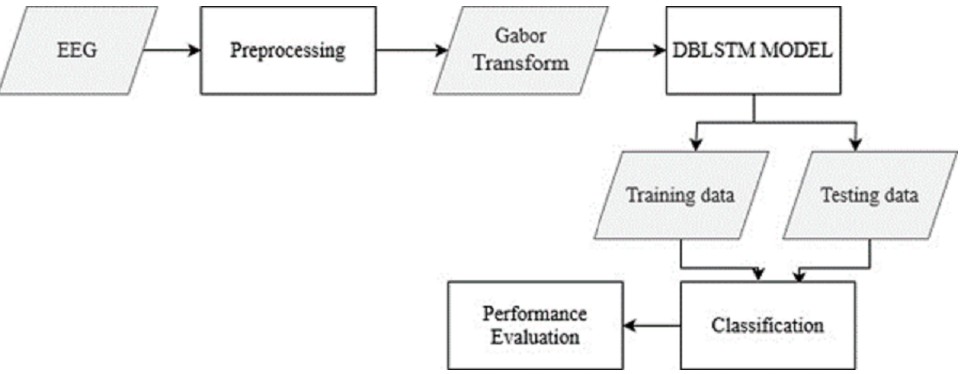

**Figure 1** The overall structure of Parkinson's disease detection methodology.

presents a high-level overview of the method, illustrating the flow and relationship between the various stages of the automated identification of Parkinson's disease using deep learning and EEG analysis.

Filtering techniques play a crucial role in preprocessing EEG signals. Bandpass filtering is commonly applied to EEG data, as these signals encompass a frequency range of 0.5 Hz to 100 Hz or even more. This process effectively mitigates unwanted low-frequency drift and high-frequency noise, ensuring that the underlying neural activity remains prominent. Additionally, notch filtering is employed to combat power line interference, typically occurring at 50 Hz or 60 Hz, depending on the geographical region. In parallel, artifact removal techniques are instrumental in refining EEG data quality. Independent Component Analysis (ICA) is a powerful tool for blind source separation, effectively isolating EEG signals into independent components. This separation aids in identifying and removing artifacts like eye blinks and muscle activity from the EEG dataset.

The proposed deep-learning model, DLBLSTM, is introduced. This architecture is specifically designed to address the challenges posed by the non-linear, non-stationary, and complex nature of EEG signals. DLBLSTM represents each layer as the sum of its hidden state and the hidden states of all layers above it. This representation is recursively transmitted to all layers below it, allowing for a rich and comprehensive representation of the EEG data. The DLBLSTM model is trained using the spectrograms obtained from the Gabor transform as input data. The model learns the underlying patterns and features distinguishing PD-positive and PD-negative EEG recordings. During training, methods like backpropagation and gradient descent are used to fine-tune the settings of the model. Once the DLBLSTM model is trained, it can classify new EEG recordings as PD-positive or PD-negative. The model takes the spectrogram of the input EEG signal as input and produces a classification output based on the learned representation of PD-related patterns. A robust sixfold cross-validation method measures, how well the suggested deep-learning model performs. The dataset is partitioned into six parts; five are used to train the model, while the sixth is used to assess its efficacy. This procedure has six iterations, each using a distinct portion of the data for validation. The classification accuracy is computed to measure how well the model identifies Parkinson's disease. Cross-validation with more

prominent folds (*e.g.*, 10-fold) provides a more accurate estimate of model performance but can be computationally expensive, especially for complex models or large datasets. Six-fold cross-validation strikes a balance by offering a reasonable assessment of model performance while being less computationally intensive than methods with more folds.

## Dataset

The open neuro dataset comprises 15 PD patients and 16 HC controls. Its origin may be traced back to the University of California, (*Rockhill et al., 2020*). Both groups had similarly aged and sexed participants who scored again on cognitive tests and were equally likely to be right-handed. The average duration of their condition was 4.5 to 3.5 years. The data was collected using a counterbalanced order of PD patients taking or not taking their medication. During OFF medication recordings, HC individuals went without medication for at least 12 h, whereas ON medication recordings involved regular dosing. We sampled at 512 Hz using a 32-channel Bio semi-active Two EEG device. At least 3 min of resting state EEG data was obtained and pre-processed. Muscle activity artifacts, electrical noise, eye blinks and motions, and other sounds have all been evaluated and eliminated by hand. The 0.5 Hz high pass filter has been helpful to all EEG replays.

## Gabor transform

The Gabor transform is a mathematical technique used to analyze signals in the time-frequency domain. It involves convolving a signal with a set of Gabor functions, which are complex exponential-modulated Gaussian functions. The Gabor transform of a signal can be obtained by performing a convolution operation in both the time and frequency fields. The Gabor transform, specifically the Gabor Wavelet Transform, is a valuable tool in signal processing, including analyzing EEG (Electroencephalogram) signals. One of the primary advantages of the Gabor transform is its ability to provide time-frequency localization of signal components. This means it can reveal how the frequency content of a signal evolves. EEG signals are often non-stationary, meaning their frequency characteristics change over time. The Gabor transform can capture these changes accurately. The Gabor transform strikes a good balance between time and frequency resolution, making it suitable for analyzing signals with both high and low-frequency components. This is important in EEG analysis, as EEG signals often contain frequencies associated with different brain activities. While the Gabor transform offers many advantages, other time-frequency analysis techniques are commonly used for EEG signal analysis, including Short-Time Fourier Transform (STFT) and Continuous Wavelet Transform (CWT).

The Gabor transform of a continuous-time signal x(t) is specified by the subsequent equation

$$X(t,f) = \int_{-\infty}^{\infty} X(\tau)g(t-\tau,f)e^{-j2\pi f\tau}d\tau \tag{2}$$

where $X(t,f)$ represents the Gabor transform of the signal $x(t)$ at time $t$ and frequency $f$. The function $g(t,f)$ represents the Gabor kernel, given by:

$$g(t,f) = e^{-\frac{t^2}{2\sigma_t^2}} e^{j2\pi ft} \tag{3}$$

Here, $\sigma_t$ represents the standard deviation of the Gaussian window used in the Gabor transform. The term $e^{-\frac{t^2}{2\sigma_t^2}}$ represents the temporal windowing function, which localizes the analysis in the time domain. In contrast, $e^{j2\pi ft}$ represents the complex exponential modulation that localizes the study in the frequency domain. In practice, the Gabor transform is often applied to discrete-time signals. The discrete Gabor transform can be computed using a discrete version of the convolution operation, and the Gabor kernel can be discretized accordingly. The Gabor transform is commonly used in EEG signal processing to convert raw EEG data into time-frequency representations such as spectrograms. Applying the Gabor transform to EEG recordings allows the time-varying spectral content of the signals to be analyzed, enabling the extraction of relevant features for further analysis and classification tasks.

## Densely linked bi directional LSTM (DLBLSTM)

Detecting PD from EEG signals is a challenging task requiring sophisticated machine-learning models. Dense connected Bi-LSTM (long short-term memory) architecture is one approach that can be used for this purpose. Combining Bi-LSTM and dense connections can help capture both temporal dependencies and complex patterns in the EEG data. RNNs, particularly LSTM networks, have been widely employed in many sophisticated neural networks for training to categorize, manipulate, and forecast time series because of their capacity to learn dependence over time recurrently. We will first quickly go through LSTM and its expansions, then introduce the new DLBLSTM we suggested.

### *Long short-term memory*

LSTM is an RNN architecture intended to process and model sequential data. LSTM networks excel in collecting long-range relationships in time-series information, making them a good choice for sequence-based applications like voice recognition, NLP, and interval forecasting. The critical innovation of LSTM networks is shown in the Fig. 2 lies in their ability to maintain a memory state over time, allowing them to forget or retain information from previous time steps selectively. This is achieved through unique gating mechanisms that control the flow of information within the network.

Cell state $(Ct)$ acts as the memory of the LSTM and runs along the entire sequence. It can selectively learn to recall or forget data over time. This continuity of information flow is what enables LSTMs to maintain long-range dependencies.

Hidden state $(ht)$ at each time step acts as the LSTM cell's output and serves as the input to the next time step. It carries relevant information learned from previous time steps. Input Gate $(i)$, Forget Gate $(f)$, and Output Gate $(o)$ are utilized to manage how data enters and leaves the LSTM cell. There are three gates: the input gate, the forget gate, and the gate that outputs the result. The input gate supervises the amount of data supplied to the cell state, the forget gate decides what data is discarded from the cell state, and the output gate governs how much knowledge leaves and enters the hidden state.

The equations governing the LSTM cell are as follows:

$$i(t) = sigmoid(Wi * [ht - 1, xt] + bi) \tag{4}$$

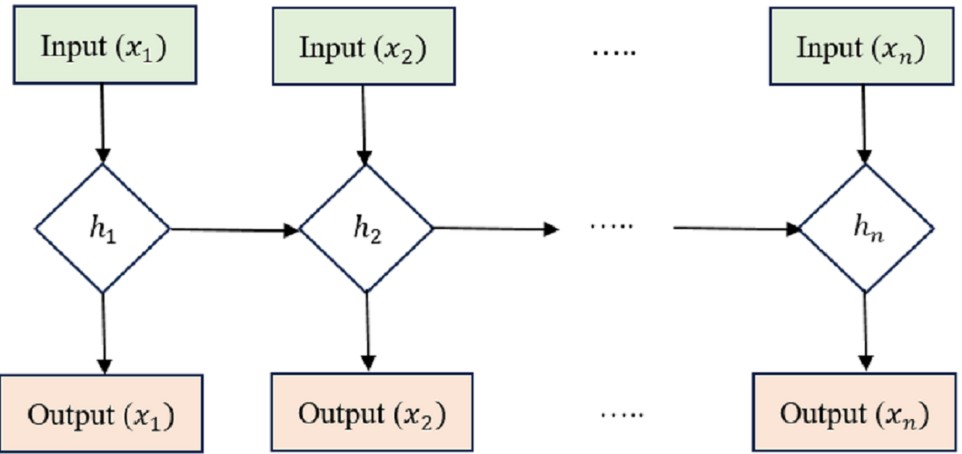

**Figure 2** Long short-term memory structure.

$$f(t) = sigmoid(Wf * [ht-1, xt] + bf) \tag{5}$$

$$o(t) = sigmoid(Wo * [ht-1, xt] + bo) \tag{6}$$

$$\hat{C}t = tanh(Wc * [ht-1, xt] + bc) \tag{7}$$

$$Ct = f(t) * Ct-1 + i(t) * \hat{C}t \tag{8}$$

$$ht = o(t) * tanh(Ct) \tag{9}$$

In these equations, $xt$ isthe input at time $t$, and the variables with 'W' and 'b' represent the learnable weights and biases of the LSTM cell. LSTMs have proven very effective in handling vanishing and exploding gradient problems, which are common issues in training traditional RNNs on long sequences. Using gating mechanisms, LSTMs can selectively retain important information and mitigate the vanishing gradient problem, making them a powerful tool for sequential data processing. Table 1 presents the layered architecture of the long short-term memory (LSTM) model employed in our research. This architecture comprises several key layers, each with distinct roles in the model's overall functioning.

The first layer in our model is an LSTM layer. LSTM units are essential components of recurrent neural networks, known for their ability to capture long-range dependencies in sequential data. In this layer, the output shape is specified as (None, 1, 64), indicating that it generates sequences with a length of 1 and a feature dimension of 64. The parameter

**Table 1  LSTM model layered architecture.**

| Layer (type) | Output shape | Param |
|---|---|---|
| lstm (LSTM) | (None, 1, 64) | 668,928 |
| dropout (Dropout) | (None, 1, 64) | 0 |
| $lstm_1$ (*LSTM*) | (None, 32) | 12,416 |
| $dropout_1$ (*Dropout*) | (None, 32) | 0 |
| dense (Dense) | (None, 2) | 99 |

count for this layer is 668,928, reflecting the weights and biases learned during training. Following the LSTM layer, we have a dropout layer. Dropout is a regularization technique that helps prevent overfitting by randomly setting a fraction of input units to zero during each forward pass. The second LSTM layer, labeled as $lstm_1$, is designed to reduce the dimensionality of the sequence output. It transforms the sequence with a length of 1 into a single vector of 32. Like the previous *dropoutlayer*, $dropout_1$ is applied to the output of $lstm_1$. It maintains an output shape of ($None, 32$). The final layer in our LSTM model is a dense layer, which is a fully connected layer. It takes the output of the previous layer and produces a final prediction. In this case, the output shape is ($None, 2$), which outputs a vector of length 2.

### Bi-directional LSTM

The LSTM has been expanded into the bi-directional LSTM (Bi-LSTM), which can process requests in both the forward and backward directions. This makes the Bi-LSTM more efficient in collecting contextual and relationships in historical data, as it can take into account details from both previous times and the near future.

The equations for a Bi-LSTM cell, as shown in Fig. 3, combine the forward LSTM and backward LSTM operations. For forwarding LSTM,

Input Gate ($i_f$):

$$i_f(t) = sigmoid(Wi_f * [h_f(t-1), x(t)] + bi_f) \tag{10}$$

Forget Gate ($f_f$):

$$f_f(t) = sigmoid(Wf_f * [h_f(t-1), x(t)] + bf_f) \tag{11}$$

Output Gate ($o_f$):

$$o_f(t) = sigmoid(Wo_f * [h_f(t-1), x(t)] + bo_f) \tag{12}$$

Candidate Cell State ($\hat{C}_f$):

$$\hat{C}_f(t) = tanh(WC_f * [h_f(t-1), x(t)] + bC_f) \tag{13}$$

Cell State ($C_f$) Update:

$$C_f(t) = f_f(t) * C_f(t-1) + i_f(t) * \hat{C}_f(t) \tag{14}$$

Hidden State ($h_f$):

$$h_f(t) = o_f(t) * tanh(C_f(t)) \tag{15}$$

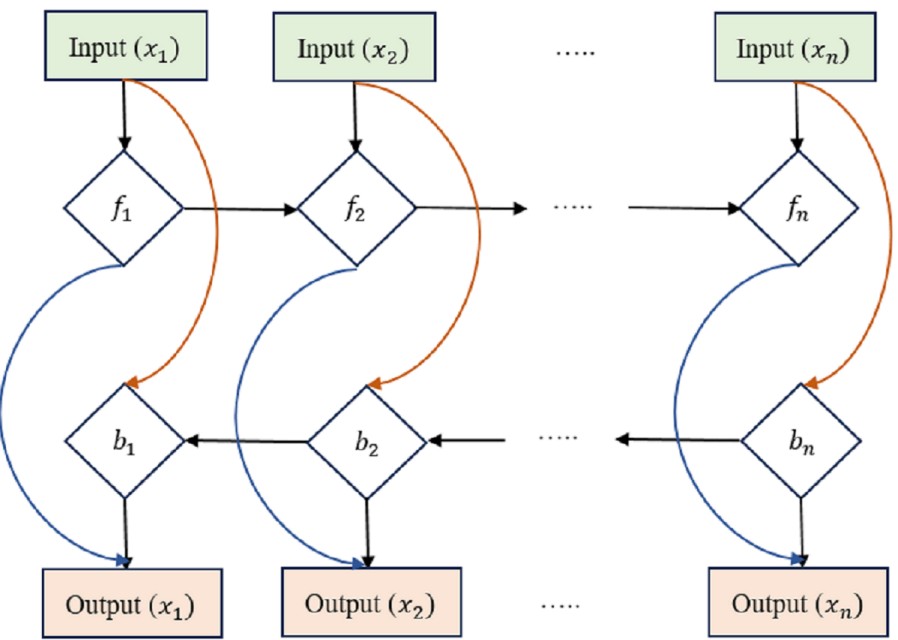

**Figure 3  Bi-directional long short-term memory structure.**

where $h_f(t)$ is the hidden state of the forward LSTM at time step $t$, $x(t)$ is the input at time $t$, and the variables with 'W' and 'b' represent the learnable weights and biases of the forward LSTM. For backward LSTM,

Input Gate ($i_b$):

$$i_b(t) = sigmoid(Wi_b * [h_b(t+1), x(t)] + bi_b) \tag{16}$$

Forget Gate ($f_b$):

$$f_b(t) = sigmoid(Wf_b * [h_b(t+1), x(t)] + bf_b) \tag{17}$$

Output Gate (o_b):

$$o_b(t) = sigmoid(Wo_b * [h_b(t+1), x(t)] + bo_b) \tag{18}$$

Candidate Cell State ($\hat{C}_b$):

$$\hat{C}_b(t) = tanh(Wc_b * [h_b(t+1), x(t)] + bc_b) \tag{19}$$

Cell State ($C_b$) Update:

$$C_b(t) = f_b(t) * C_b(t+1) + i_b(t) * \hat{C}_b(t) \tag{20}$$

Hidden State ($h_b$):

$$h_b(t) = o_b(t) * tanh(C_b(t)) \tag{21}$$

where $h_b(t)$ is the hidden state of the backward LSTM at time step $t$, $x(t)$ is the input at time $t$, and the variables with 'W' and 'b' represent the learnable weights and biases of the

**Table 2** Layered architecture of DLBLSTM.

| Layer (type) | Output shape | Param |
|---|---|---|
| inputs$_l$ stm (*InputLayer*) | (None, 178, 1) | 0 |
| Dense (Dense) | (None, 178, 32) | 64 |
| Bidirectional (Bidirectional) | (None, 256) | 1, 64, 864 |
| Dropout (Dropout) | (None, 256) | 0 |
| batch$_n$ ormalization (*BatchNo*) | (None, 256) | 1, 024 |
| dense$_1$ (*Dense*) | (None, 64) | 16, 448 |
| dropout$_2$ (*Dropout*) | (None, 64) | 0 |
| batch$_n$ ormalization$_1$ (*BatchNo*) | (None, 64) | 256 |
| dense$_3$ (*Dense*) | (None, 2) | 130 |

backward LSTM. Bi-LSTMs often combine forward and reverse hidden states into a single output at each time step t:

$$h_b i(t) = [h_f(t), h_b(t)] \tag{22}$$

The architecture of the neural network model, as shown in Table 2, is delineated through a series of interconnected layers, each with specific characteristics. The input layer (*inputs$_l$stm*) serves as the point of entry, specifying the input data's expected shape, consisting of sequences with a length of 178 and a feature dimension of 1. Subsequently, the dense layer (*Dense*) applies a linear transformation to the input, generating an output sequence with sizes (*None*, 178, 32). This transformation involves 64 trainable parameters. The bidirectional layer (*Bidirectional*) is a pivotal component, enveloping two LSTM layers to capture bidirectional dependencies within the input data. As a result, it yields an output vector of dimension 256, embedding 164, 864 trainable parameters. To prevent overfitting, the dropout layer (*Dropout*) is employed, randomly setting a portion of input units to zero during forward passes without any trainable parameters. The batch normalization layer (*batch$_n$ormalization*) comes into play, normalizing activations to enhance training stability, with 1, 024 trainable parameters for scaling and shifting. Following this, the dense layer (*dense$_1$*) further reduces the data's dimensionality to 64 dimensions, incorporating 16, 448 trainable parameters. Subsequently, the dropout layer (*dropout$_2$*) reprises its role in regularization, with no trainable parameters, and is succeeded by the batch normalization layer (*batch$_n$ormalization$_1$*) with 256 trainable parameters for scaling and shifting. Lastly, the final output is generated by the dense layer (*dense$_3$*), producing a vector of length 2, signifying the model's ultimate prediction. This layer comprises 130 trainable parameters. These layers constitute a complex neural network architecture designed to process sequential data, capture bidirectional information, and produce meaningful predictions while mitigating overfitting through dropout and stabilizing training with batch normalization.

### Densely linked Bi-LSTM

We present an early version of the densely-connected LSTM network here, which is based on the idea of densely linked networks. Incorporating skip connections into the densely

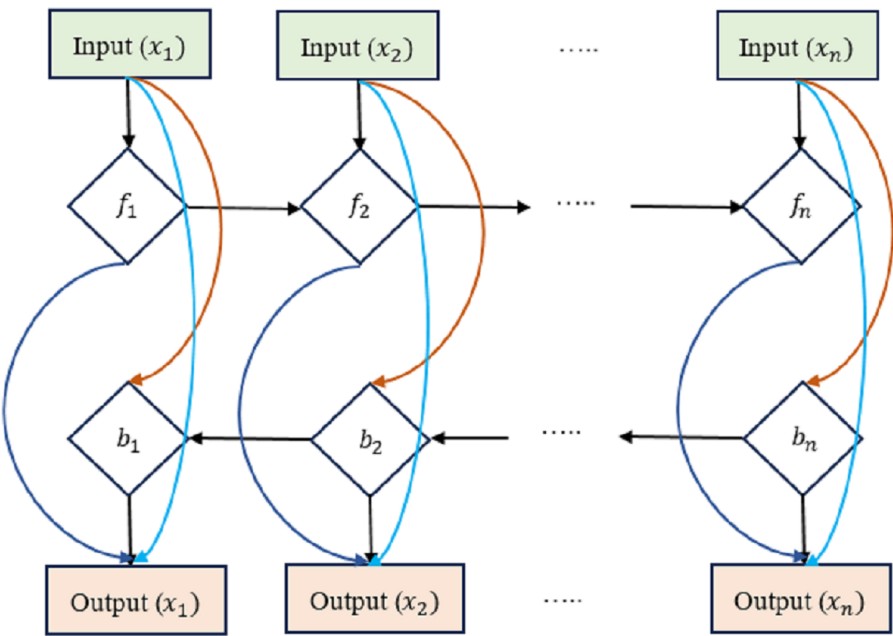

**Figure 4** Densely-linked bidirectional LSTM (DLBLSTM) networks.

linked network nodes is fundamental to our method. Figure 4 depicts this architecture, with $d_{lt}$ standing for the densely linked hidden unit of the $l^{th}$ layer at the $t^{th}$ time step. In this research, the strongly interconnected layers show the skip-connections between them. Connecting and integrating data between the many tiers of a network is what these lines are for. In contrast to the element-wise additive operation used by the residual learning framework, the concatenation operation is used here. This replacement counteracts the performance drop that might occur with direct gradient backpropagation. Densely-Linked Bidirectional LSTM (DLBLSTM) networks are an innovative method for modeling temporal patterns that benefit from densely linked construction and bi-directional temporal features.

The DLBLSTM architecture is unique in deep learning in that it features connections between neighboring layers and levels that are physically separated from one another. Because of its one-of-a-kind topology, the network's many groups may exchange data and communicate more efficiently. Multiple disciplines of study have documented and investigated information transmission phenomena in both directions simultaneously. The possible ramifications of this quality, bidirectional information flow, have garnered much interest. Authors have studied the mechanics and characteristics of two-way communication. The two-way transmission of information has been investigated through experiments and theoretical models. It has been shown that DBLSTM's architecture eliminates gradient vanishing, boosts feature transmission, promotes the reuse of features, and significantly lowers the number of variables. It improves the representation of spatial and short-term temporal features. Two LSTM networks are used to create the DB-LSTM model's two-layer architecture. Densely interwoven skip-connections link together these

LSTM networks. LSTM connections, one running forward and one running reversed, can simulate the long-term temporal trends in operations. All of the results from each iteration are merged into one final result. Each DB-LSTM unit is definite as follows:

$$\overleftrightarrow{d_t} = \left[ \overleftarrow{d_t} , \overrightarrow{d_t} \right] \tag{23}$$

where $\overleftrightarrow{d_t}$ is the $t$-th result from DB-LSTM. When LSTM networks are paired with dense skip-connections, the forward and backward directional outputs at the $t$th time step are denoted by $\overrightarrow{d_t}$ and $\overleftarrow{d_t}$, correspondingly. Concatenation is represented by the letter [, ]. The directions $\leftarrow$ and $\rightarrow$ of the output d are indicative of the forward and backward directions of the input patterns. The following defines the influence of preceding layers on the output of the $l$th layer of the LSTM block $d_t^l$ at the $t$th time step:

$$d_t^l = \left[ h_t^l \left( \left[ d_t^0, d_t^2, \ldots\ldots d_t^{l-1} \right] \right), x_t \right] \tag{24}$$

where $\left[ d_t^0, d_t^2, \ldots\ldots d_t^{l-1} \right]$ is the sum of the characteristics retrieved from the preceding levels.

The LSTM layer's input is $X$, and it receives features from the previous $t$ time steps.

$$\left[ h_t^l \left( \left[ d_t^0, d_t^2, \ldots\ldots d_t^{l-1} \right] \right), x_t \right]$$

represents the output of the previous *LSTM* layer plus the input feature $x_t$ at the $t^{th}$ time step. The lth LSTM layer, during the tenth time step, is denoted by $h_t^l$. The cross-entropy formula is used to determine the amount of damage.

$$\psi(y, \varphi) = -\sum_{k=1}^{N} y_i \left( \varphi_i - \log \sum_{k=1}^{N} exp \; \varphi_i \right) \tag{25}$$

where $N$ is the total number of categories and $y_i$ is the $i$th video's label. The fusion layer now includes the scores generated by long-term temporal modeling. A multi-scale sliding window combines the scores at the end of each time step.

$$w_q = \frac{1}{(T-p)p} \sum_{n=1}^{N-p} \sum_{m=n}^{n+p} E_t^{bi} \tag{26}$$

where $T$ is the total number of time steps (also known as segments), and $n$ is the initial time step at which the sliding window will begin to operate. The typical synthesis of multi-scale sliding windows addresses the issue of events occurring at various times.

Table 3 is mentioned about the layered architecture of Bi-LSTM. The architectural composition of the DLBLSTM model is delineated through a sequence of interconnected layers, each serving a specific purpose within the neural network's framework. The initial layer, the input layer ($inputs_lstm$), acts as the point of entry for the model, anticipating input data structured as sequences with a length of 178 and a feature dimension of 1. Following this, the dense layer (dense) undertakes a linear transformation of the input data, resulting in an output shape of ($None, 178, 32$) and contributing 64 trainable parameters to the model. The subsequent bidirectional layer is the centerpiece of the architecture, encapsulating two

**Table 3  Layered architecture of Bi-LSTM.**

| Layer (type) | Output shape | Param |
|---|---|---|
| inputs$_l$ stm (*InputLayer*) | (None, 178, 1) | 0 |
| Dense (Dense) | (None, 178, 32) | 64 |
| Bidirectional (Bidirectional) | (None, 178, 64) | 16640 |
| Dropout (Dropout) | (None, 178, 64) | 0 |
| dense$_1$ (*Dense*) | (None, 178, 32) | 2080 |
| bidirectional$_1$ (*Bidirectional*) | (None, 178, 64) | 16640 |
| dropout$_1$ (*Dropout*) | (None, 178, 64) | 0 |
| batch$_n$ ormalization (*BatchNorm*) | (None, 178, 64) | 256 |
| dense$_2$ (*Dense*) | (None, 178, 2) | 130 |

LSTM layers that process input sequences bidirectionally. This operation yields an output shape of $(None, 178, 64)$ and encompasses $16,640$ trainable parameters.

The dropout layer (dropout) promotes model generalization, randomly deactivating a fraction of input units during each forward pass. Importantly, it introduces no trainable parameters. The ensuing dense layer ($dense_1$) reduces feature dimensionality to $(None, 178, 32)$ while incorporating $2,080$ trainable parameters. The second bidirectional layer ($bidirectional_1$) mirrors the operation of its predecessor, processing data bidirectionally and yielding an output shape of $(None, 178, 64)$. This layer adds another $16,640$ trainable parameters to the model.

A second dropout layer ($dropout_1$) follows, enhancing regularization without introducing trainable parameters. To stabilize training and facilitate convergence, the batch normalization layer ($batch_normalization$) is incorporated, normalizing activations from the prior layer and enhancing training stability by introducing 256 trainable parameters for scaling and shifting.

Lastly, the final dense layer ($dense_2$) generates the model's output, producing a sequence of length 178 with two features, denoted as $(None, 178, 2)$. This layer encompasses 130 trainable parameters. This DLBLSTM architecture is meticulously designed to effectively process sequential data, capturing past and future context through bidirectional LSTM layers while maintaining regularization and stability through dropout and batch normalization layers, ultimately leading to meaningful predictions based on input sequences.

In Table 4, each hyperparameter is listed. The "Values to Try" column specifies the different values or options you can experiment with during hyperparameter tuning. The "Best Value" column indicates the best-performing value or setting for each hyperparameter based on your validation results.

## RESULT AND DISCUSSION

Implementing a densely linked bidirectional long short-term memory (DLBLSTM) model in Python typically involves deep learning libraries such as TensorFlow or PyTorch.

The train-test-validation split of a dataset, commonly referred to as an 80-10-10 split, is used approach in this research for partitioning data into three distinct subsets. In this

**Table 4  Hyper parameter tuning.**

| Hyperparameter | Values to try | Best value |
|---|---|---|
| Number of LSTM layers | 1, 2, 3 | 2 |
| LSTM Units | 32, 64, 128 | 64 |
| Bidirectional | True, False | True |
| Dropout rate | 0.2, 0.4, 0.6 | 0.4 |
| Learning rate | 0.001, 0.01, 0.1 | 0.001 |
| Batch size | 32, 64, 128 | 64 |
| Epochs | 10, 20, 30 | 20 |
| Activation function | 'relu', 'tanh', 'sigmoid' | 'relu' |
| Loss function | 'categorical$_c$ rossentropy', 'MSE' | 'categorical$_c$ rossentropy' |

setup, 80% of the data is allocated to the training set, 10% to the test set, and another 10% to the validation set. The training set, comprising the largest portion of the data, plays a pivotal role in training the machine learning model. During the training phase, the model learns patterns, features, and relationships within the data, which enables it to make predictions or classifications. It is essentially the foundation upon which the model is built. The test set, representing 10% of the data, is reserved for evaluating the model's performance after training. It serves as an independent dataset that the model has never seen during its training phase. By assessing the model's accuracy, precision, recall, or other relevant metrics on the test set, we gain insights into how well the model generalizes to new, unseen data. This step is crucial for assessing the model's real-world applicability and detecting potential overfitting. The validation set, also consisting of 10% of the data, acts as an intermediary between training and testing. It helps in tuning hyperparameters, such as learning rates or regularization strengths, without contaminating the test set. By evaluating the model's performance on the validation set during training, adjustments can be made to optimize the model's architecture and parameters, leading to better generalization on the test set and, ultimately, improved model performance.

To assess the performance of PD detection using the DLBLSTM model, several standard evaluation metrics can be used. These measures reveal how well the model distinguishes between Parkinson's and other diseases. Precision is the ratio of accurately forecast remarks to the total number of examples, and it is an indication of how well the model performs when making forecasts.

Regularization methods constrain the model's optimization process, discouraging it from fitting the training data too closely. L1 and L2 regularization techniques add a penalty term to the loss function based on the model's weights' absolute (L1) or squared (L2) values. They encourage weight values to be small. Dropout randomly deactivates a fraction of neurons during each training batch. It prevents the model from relying too heavily on specific neurons and promotes generalization. They are implementing early stopping by monitoring the model's performance on a validation dataset during training. If the routine begins to degrade (*e.g.*, validation loss increases), stop training to prevent overfitting. Accuracy is the ratio of correct Parkinson's disease identifications (positive forecasts) to the total number of positive forecasts. When a model has high precision, it means it

**Table 5**  **Performance result of the proposed model DLBLSTM.**

| Performance metrics | Normal | PD |
|---|---|---|
| Accuracy | 0.99 | 1.00 |
| Sensitivity/Recall | 0.99 | 0.99 |
| Specificity | 0.99 | 1.00 |
| Precision | 1.00 | 0.99 |
| F1-Score | 0.99 | 0.99 |

produces a few erroneous predictions.

$$Accuracy(A) = \frac{T_{pt} + T_{nt}}{T_{pt} + T_{nt} + F_{pt} + F_{nt}} \tag{27}$$

Known also as "sensitivity" or "true positive rate", recall enumerates the rate at which positive instances of Parkinson's disease are correctly forecast qualified to the total number of positive samples.

$$Precision(P) = \frac{T_{pt}}{T_{pt} + F_{pt}} \tag{28}$$

$$Recall(R) = \frac{T_{pt}}{T_{pt} + F_{nt}} \tag{29}$$

The F1-score is the arithmetic mean of recall and accuracy. In the case of unbalanced datasets, it offers a fair assessment metric.

$$F1 - score(F1) = 2 \times \frac{P \times R}{P + R}. \tag{30}$$

The performance evaluation results for the proposed DLBLSTM model in PD detection are exposed in Table 5. For the "Normal" class (representing individuals without Parkinson's disease), the model demonstrates highly accurate and reliable predictions. The accuracy is 0.99 (99%), meaning that 99% of the instances in the "Normal" class were correctly classified. Sensitivity (also known as Recall) is also 0.99 (99%), indicating that the model correctly identified 99% of the individuals without Parkinson's disease from the total number of actual "Normal" class instances. Specificity, which represents the capability of the model to identify negative instances correctly, is also 0.99 (99%), implying that the model appropriately recognized 99% of the individuals without Parkinson's disease out of all the actual negative instances. The precision for the "Normal" class is 1.00 (100%), indicating that all the instances classified as "Normal" by the model were indeed true "Normal" class instances. The F1-score is the average of the recall and accuracy scores, is 0.99 (99%), reflecting the balanced performance of the model in the "Normal" class.

For the "PD" class (representing individuals with Parkinson's disease), the DLBLSTM model achieves exceptional performance. The accuracy is 1.00 (100%), indicating that all instances in the "PD" class were correctly classified. The sensitivity/recall for the "PD" class is 0.99 (99%), showing that the model correctly identified 99% of the individuals with

Parkinson's disease from the total number of actual ''PD'' class instances. The specificity is 1.00 (100%), signifying that the model correctly identified all negative instances (individuals without Parkinson's disease) from the total number of negative examples. The precision for the ''PD'' class is 0.99 (99%), indicating that 99% of the instances classified as ''PD'' by the model were indeed true ''PD'' class instances. The F1-score for the ''PD'' class is 0.99 (99%), confirming the balanced performance of the model in detecting individuals with Parkinson's disease.

The accuracy and recall sum metric (AUC-ROC) measures how well a model performs across a range of cutoffs. The area under the receiver operating characteristic (AUC-ROC) curve has a greater value if the true positive rate is higher than the false positive rate. The confusion matrix breaks down the model's predictions in great depth. Predictions are displayed as either true positives ($T_{pt}$), true negatives ($T_{nt}$), false positives ($F_{pt}$), or false negatives ($F_{nt}$). To visualize the model's performance, we may plot the true positive rate *vs.* the false positive rate at various classification levels to create an ROC curve.

Figure 5 refers to a visual representation of the performance evaluation results for the Parkinson's disease detection model named ''DLBLSTM''. The reported accuracy values indicate how well the model performs on the training and testing datasets. The training accuracy of 100% suggests that the DLBLSTM model achieved perfect accuracy on the data it was trained on. This means that during the training process, the model could appropriately predict the labels of all the training samples, leaving no errors in the training set. The testing accuracy of 99.6% indicates the performance of the DLBLSTM model on an unseen or ''out-of-sample'' dataset, which is used to assess the model's generalization ability. The model achieved an accuracy of 99.6% on this separate testing dataset, meaning it correctly predicted the labels of approximately 99.6% of the samples in the test set. It is significant to communicate that while the training accuracy of 100% might suggest that the model is performing perfectly, it is essential to interpret this cautiously. A training accuracy of 100% can indicate potential overfitting, where the model has memorized the training data without generalizing well to new, unseen data. This is why the testing accuracy is crucial, as it provides a more realistic assessment of the model's performance on new data.

Rendering to the findings, a low training loss value, specifically 0.02, indicates that the model is effectively learning from the training data, as shown in Fig. 6. This implies that the model can generate precise predictions based on the information it has been exposed. To assess the model's generalization capability, it is crucial to evaluate the testing loss, which in this particular instance is recorded as 0.03. This evaluation determines whether the model is prone to overfitting, a phenomenon that should be avoided. Overfitting is a phenomenon observed in machine learning models where the model achieves high performance on the training data but fails to generalize well to new, unnoticed data. This is typically indicated by a higher testing loss, which measures the model's presentation of the unseen data, compared to the training loss, which procedures the model's performance on the training data.

Using a confusion matrix is a common practice in measuring the effectiveness of a classification model. It involves the construction of a table that facilitates the evaluation process. The present analysis provides a complete overview of the model's predictive

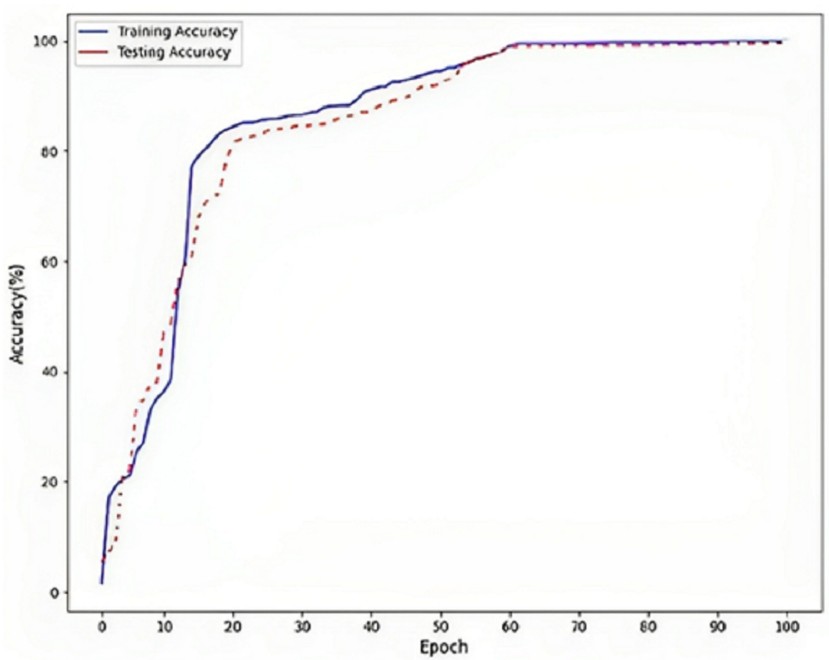

**Figure 5** Accuracy on DLBLSTM model on Parkinson's disease detection.

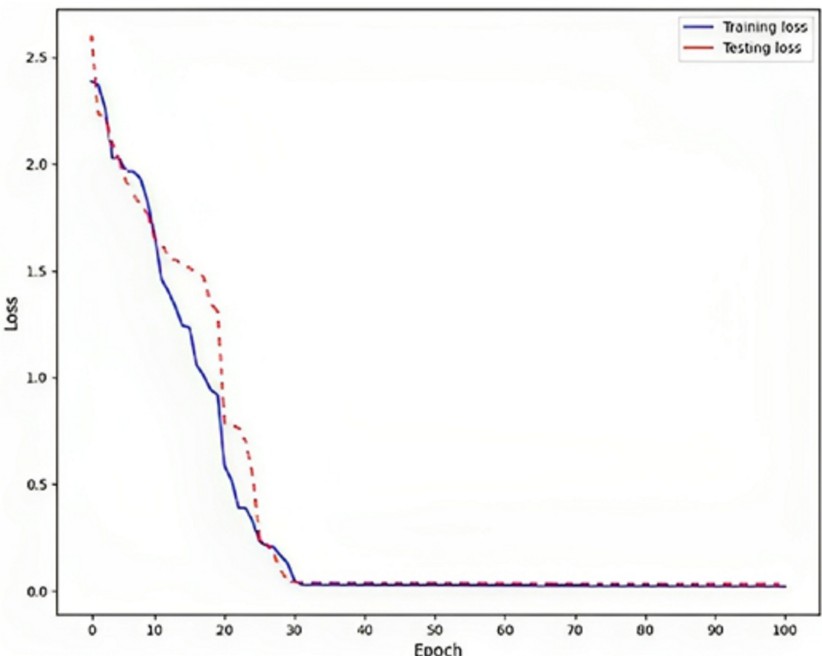

**Figure 6** Loss on DLBLSTM model on Parkinson's disease detection.

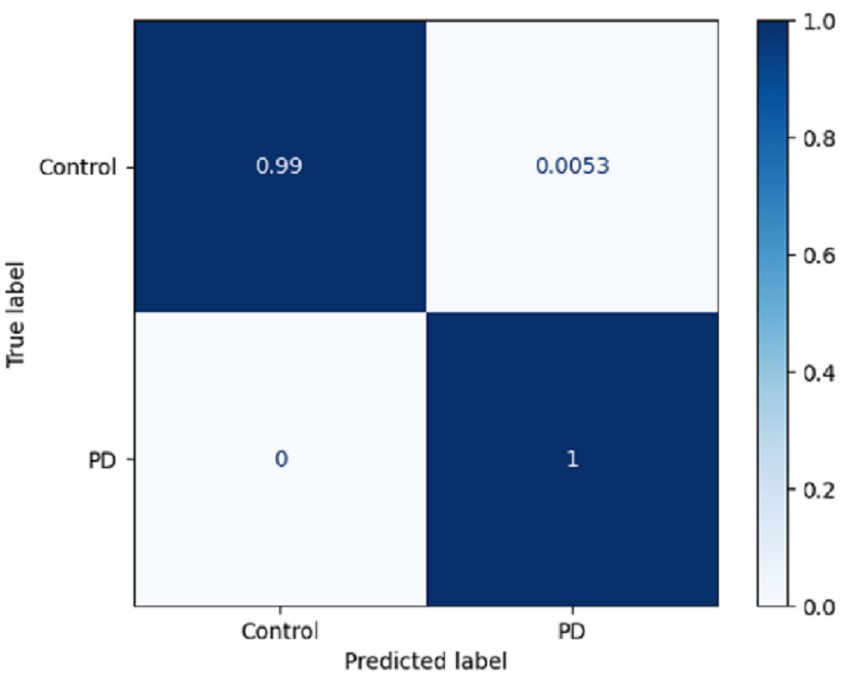

**Figure 7  Confusion matrix on DLBLSTM model on Parkinson's disease detection.**

performance about the actual ground truth across various classes. Figure 7 presents a graphical representation or chart showcasing the evaluation metrics associated with the DLBLSTM model for the recognition of PD. In machine learning and classification tasks, a confusion matrix is a pivotal assessment tool, thoughtfully structured as a table housing four distinct values. These values are the true positives (TP), which signify the instances where the model accurately forecasts the positive class; the true negatives (TN), representative of instances correctly predicted as the negative class; and the false positives (FP), denoting those unfortunate occasions when the model erroneously predicts the positive class, constituting a Type I error. The false negatives (FN) are equally significant, capturing instances where the model incorrectly anticipates the negative class, embodying a Type II error. This structured matrix offers a comprehensive view of a classification model's performance, enabling the evaluation of its efficacy in distinguishing between different classes or labels in binary or multiclass classification scenarios. Normalization is achieved by dividing the counts in the confusion matrix by various factors, typically the sum of counts in a row or column.

The true negative rate, known as the specificity, is reported to be 0.99 in the DLBLSTM model. This indicates that the model accurately classified 99% of the instances belonging to the control class as control. The true positive rate, also known as sensitivity or recall, is a performance metric used to evaluate the DLBLSTM model's capacity to accurately label cases as PD when they correspond to the PD class. In this case, the true positive rate is 1.00, indicating that the model achieved a perfect accuracy of 100% in correctly identifying all instances from the Parkinson's disease class as PD. The true positive rate, also referred

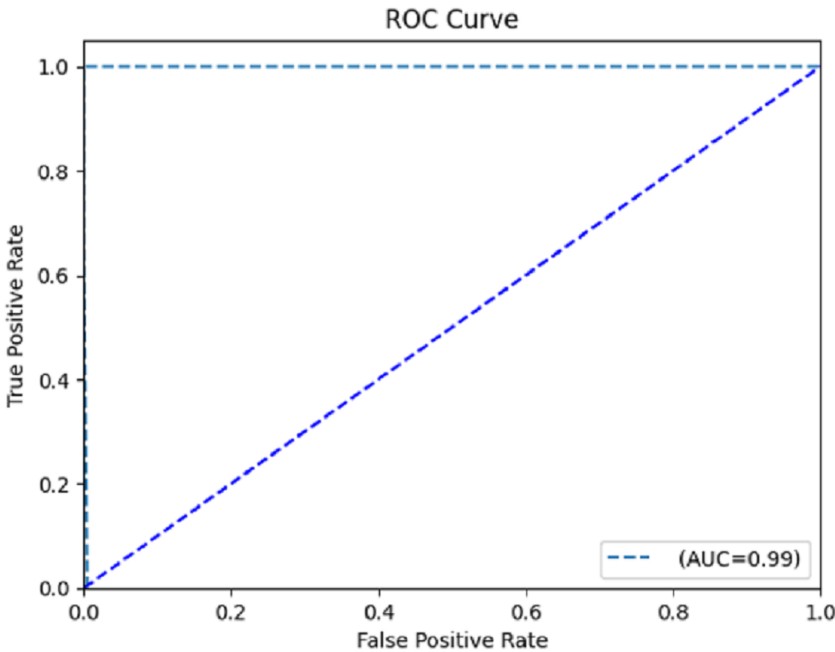

**Figure 8** Receiver operating characteristic (ROC) curve.

to as sensitivity or recall, of 1.00 for PD suggests that the model accurately identified all instances of PD within the test dataset.

The computation of the ROC curve is a common technique used to assess the performance of a network. This curve is calculated for both classes to provide a comprehensive evaluation. The calculation of the area under the curve (AUC) yielded a value of 0.996, as determined from the ROC curve depicted in Fig. 8. The trained model demonstrates satisfactory performance, as indicated by the ROC curve's ability to classify each class effectively.

To emphasize the performance of the DBLSTM, the proposed method's results are contrasted with those of baseline models run on the same dataset.

Table 6 evaluates and compares the performance of several models in the context of a classification task, with a primary emphasis on accuracy. The accuracy results for each model are as follows: the convolutional neural network (CNN) achieved an accuracy rate of 93.6%, suggesting its aptitude for feature extraction, particularly applicable to image or sequential data. The long short-term memory (LSTM) model achieved an accuracy of 95.8%, showcasing its suitability for modeling sequential data, commonly employed in tasks such as natural language processing and time series analysis. The bidirectional long short-term memory (BI-LSTM) model reached an accuracy of 96.2%, harnessing the advantages of considering both past and future context, which can be advantageous for capturing dependencies in sequential data. Impressively, the DLBLSTM model excelled with a remarkable accuracy of 99.6%, signifying its proficiency in combining bidirectional LSTM with dense connections, likely tailored to capture intricate temporal dependencies

**Table 6** Proposed model comparison against the base model.

| Model | Accuracy |
| --- | --- |
| CNN | 93.6% |
| LSTM | 95.8% |
| BI-LSTM | 96.2% |
| DLBLSTM | 99.6% |

within the dataset effectively. These results highlight the varying capabilities of each model, providing valuable insights into their performance on the classification task.

# CONCLUSION AND FUTURE WORK

The current study introduces a novel deep-learning model called densely linked bidirectional long short-term memory (DLBLSTM) for automated identification of Parkinson's disease (PD). Utilizing the Gabor transform in the model facilitates the conversion of unprocessed EEG data into spectrograms, enabling a more convenient and manageable representation of the data. The DLBLSTM model is based on the bidirectional LSTM architecture, an RNN type. In this architecture, each network layer incorporates its hidden state and the hidden states from all the layers above it. This representation is then recursively propagated to all the layers below it. The spectrograms were utilized as input data for training the planned model. The study's experimental results showcased the remarkable performance of the scheduled DLBLSTM model in detecting Parkinson's disease. By implementing a rigorous sixfold cross-validation technique, the model demonstrated exceptional performance with a classification accuracy of 99.6%. The outcomes of this study establish the model's capacity to accurately and reliably detect Parkinson's disease through automated means. While the proposed DLBLSTM model has shown promising results, several avenues exist for future research and improvement in PD detection. Expanding the dataset with a more extensive and diverse range of EEG recordings can enhance the model's generalizability and real-world applicability. Investigating methods to make the deep learning model more interpretable will aid clinicians and researchers in understanding the model's decision-making process and identifying crucial biomarkers.

## Funding

This study was funded by the Deanship of Scientific Research at King Khalid University through large group Research Project under grant number (RGP2/117/44), Princess Nourah bint Abdulrahman University Researchers Supporting Project number (PNURSP2023R203), Princess Nourah bint Abdulrahman University, Riyadh, Saudi Arabia. Research Supporting Project number(RSPD2023R787), King Saud University, Riyadh, Saudi Arabia, Prince Sattam bin Abdulaziz University project number (PSAU/2023/R/1444), and by the Future University in Egypt (FUE). The funders had

no role in study design, data collection and analysis, decision to publish, or preparation of the manuscript.

### Grant Disclosures

The following grant information was disclosed by the authors:

The Deanship of Scientific Research at King Khalid University: RGP2/117/44.

Princess Nourah bint Abdulrahman University Researchers: PNURSP2023R203.

Princess Nourah bint Abdulrahman University, Riyadh, Saudi Arabia: RSPD2023R787.

Prince Sattam bin Abdulaziz University: PSAU/2023/R/1444.

The Future University in Egypt (FUE).

### Competing Interests

The authors declare there are no competing interests.

### Author Contributions

- Marwa Obayya conceived and designed the experiments, analyzed the data, prepared figures and/or tables, and approved the final draft.
- Muhammad Kashif Saeed conceived and designed the experiments, analyzed the data, prepared figures and/or tables, and approved the final draft.
- Mashael Maashi conceived and designed the experiments, performed the computation work, prepared figures and/or tables, and approved the final draft.
- Saud S. Alotaibi performed the experiments, analyzed the data, authored or reviewed drafts of the article, and approved the final draft.
- Ahmed S. Salama performed the experiments, performed the computation work, authored or reviewed drafts of the article, and approved the final draft.
- Manar Ahmed Hamza performed the experiments, analyzed the data, performed the computation work, authored or reviewed drafts of the article, and approved the final draft.

### Data Availability

The data is available at OpenNeuro, ds002778, https://openneuro.org/datasets/ds002778/versions/1.0.2, doi: 10.18112/openneuro.ds002778.v1.0.2.

### Supplemental Information

Supplemental information for this article can be found online at http://dx.doi.org/10.7717/peerj-cs.1663#supplemental-information.

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
