# Peer review of "A novel automated Parkinson’s disease identification approach using deep learning and EEG"

_PeerJ Computer Science, doi:10.7717/peerj-cs.1663_

## Round 0.1 · original submission · Major Revisions

Based on the referee reports, I recommend a major revision of the manuscript. The author should improve the manuscript, taking carefully into account the comments of the reviewers in the reports, and resubmit the paper

**Language Note:** The review process has identified that the English language must be improved. PeerJ can provide language editing services - please contact us at copyediting@peerj.com for pricing (be sure to provide your manuscript number and title). Alternatively, you should make your own arrangements to improve the language quality and provide details in your response letter. – PeerJ Staff

·

Basic reporting

The authors of this paper present a novel deep-learning model for the automated identification of Parkinson’s disease (PD). The Gabor transform, a standard method in EEG signal processing, has been used to turn the raw data from the EEG recordings into spectrograms. In this research, the authors propose a densely linked bidirectional long short-term memory (DLBLSTM), which first represents each layer as the sum of its hidden state plus the hidden states of all layers above it, then recursively transmits that representation to all layers below it. The results obtained are well-described and convincing. Overall, this is a well-written and informative paper. Some issues which need to be addressed and clarified before the paper can be accepted for publication are:

1. Language has to be thoroughly checked. Example: What does the word recital in the abstract imply?
2. Citations should be consistent.
3. The authors should add a few more references from 2022 and 2023.

Experimental design

4. How is the pre-processing carried out?
5. Why Gabor filter? Any specific advantage? What else could have been used?
6. The statement “The average duration of their condition was 4.53.5 years” on line number 213 -214 needs correction.
7. I think the data has not been collected by the authors themselves but is taken from some dataset. The language used in the “Data Collection” section should clearly imply this.
8. Is the method used here subject-dependent or subject-independent?
9. Please explain the values in the confusion matrix. Have they been normalised?
10. Which platform has been used for implementation?
11. How is the model fine-tuned?

Validity of the findings

12. The authors are getting an accuracy of 100%. Is it the testing accuracy or the training accuracy? Did they check for overfitting?

Reviewer 2 ·

Basic reporting

- In the abstract, once the abbreviation of electroencephalography is given (in line 29), there is no need to write it in long form in the next sentence once again.
- Some words do not fully reflect the academic meaning such as recital, breakdown, etc. Instead, they may be replaced like recital -> performance, breakdown -> analysis. Please also check these words.
- I couldn't see the keywords of the study. Please add them.
- Please check the citation format of the journal. For instance, in Line 127; Shi et al. Shi et al.(2019) look strange a little bit.
- In line 214: The number 4.53.5 is not clear.
- Please check the two abbreviations: DLBLSTM and DLDBLSTM. The latter is not mentioned in the manuscript. It seems there might be a typo.

Experimental design

- In general, what is the standard procedure for diagnosing PD using EEG signals (i.e. in a non-automated way)? What are the challenges/advantages of doing this?
- Please add a paragraph at the end of the Introduction (or Related Works) section that explains your novel approach. Besides, please discuss pros / cons of existing approaches by comparing them to your work.
- Please expand and clarify your workflow in Figure 1 by separating the methods and inputs. You may use a valid flowchart notation.- Why six-fold is selected for the validation method? Is there a specific reason for choosing six?

Validity of the findings

- I think the idea you propose is really interesting and sensible. But it needs to be justified. Please compare your results with other (standard) methods such as LSTM only. I guess it won't be too hard to repeat your experiments using basic models. This is a critical step to justify how successful your proposed approach is. Thus, you may justify your contribution. Besides, make a discussion of your findings to show the superiority of your proposed model.

---

## Round 0.2 · Minor Revisions

I am happy to announce that review of your manuscript is now complete. Kindly revise the manuscript as per the reviewer suggestions and resubmit it.

·

Basic reporting

No Comment

Experimental design

No Comment

Validity of the findings

No Comment

Reviewer 2 ·

Basic reporting

1) In the abstract, once the abbreviation of electroencephalography is given (in line 29), there is no need to write it in long form in the next sentence once again.

[It is still not corrected completely! The long form of EEG is cited twice in the abstract.]

2) Some words do not fully reflect the academic meaning such as recital, breakdown, etc. Instead, they may be replaced like recital -> performance, breakdown -> analysis. Please also check these words.

[It is still not corrected! The word ‘breakdown’ is still in the abstract.]

Experimental design

1) Please expand and clarify your workflow in Figure 1 by separating the methods and inputs.

[Not done as specified! Figure 1 needs to be corrected. Data and Processes should be indicated with different notation. You may use such kind of diagram given in Figure 1 as an example.
(https://www.researchgate.net/publication/343135035_Label-free_detection_of_rare_circulating_tumor_cells_by_image_analysis_and_machine_learning/figures?lo=1)]

Validity of the findings

1) I think the idea you propose is really interesting and sensible. But it needs to be justified. Please compare your results with other (standard) methods such as LSTM only. I guess it won't be too hard to repeat your experiments using basic models. This is a critical step to justify how successful your proposed approach is. Thus, you may justify your contribution. Besides, make a discussion of your findings to show the superiority of your proposed model.

[Please give details about the baseline models. The inner architectures, names (if they are known models), data split rates (%X training, %Y testing, %Z validation etc., or # of folds if applicable). Besides, please share the related source codes.]

---

## Round 0.3 · accepted · Accept

The author has addressed the reviewer's comments properly. Thus I recommend publication of the manuscript.

Reviewer 2 ·

Basic reporting

The revisions appear to have been made. I have no further suggestions.

Experimental design

The revisions appear to have been made. I have no further suggestions.

Validity of the findings

The revisions appear to have been made. I have no further suggestions.